# Sex and Race-Related DNA Methylation Changes in Hepatocellular Carcinoma

**DOI:** 10.3390/ijms22083820

**Published:** 2021-04-07

**Authors:** Wenrui Ye, Stefan Siwko, Robert Y. L. Tsai

**Affiliations:** Institute of Biosciences and Technology, Texas A&M Health Science Center, Houston, TX 77030, USA; yew@exchange.tamu.edu (W.Y.); ssiwko@tamu.edu (S.S.)

**Keywords:** health disparity, epigenetic, epigenome, liver cancer, HCC, fatty liver, metabolic disorder, gender, ethnicity

## Abstract

Hepatocellular carcinoma (HCC) is the sixth most common cancer and fourth leading cause of cancer-related death worldwide. The number of HCC cases continues to rise despite advances in screening and therapeutic inventions. More importantly, HCC poses two major health disparity issues. First, HCC occurs more commonly in men than women. Second, with the global increase in non-alcoholic fatty liver diseases (NAFLD), it has also become evident that HCC is more prevalent in some races and/or ethnic groups compared to others, depending on its predisposing etiology. Most studies on HCC in the past have been focused on genetic factors as the driving force for HCC development, and the results revealed that genetic mutations associated with HCC are often heterogeneous and involve multiple pathogenic pathways. An emerging new research field is epigenetics, in which gene expression is modified without altering DNA sequences. In this article, we focus on reviewing current knowledge on HCC-related DNA methylation changes that show disparities among different sexes or different racial/ethnic groups, in an effort to establish a point of departure for resolving the broader issue of health disparities in gastrointestinal malignancies using cutting-edge epigenetic approaches.

## 1. Health Disparity Is a Common Feature of Gastrointestinal Cancers

Gastrointestinal (GI) cancers include some of the most common cancers worldwide, such as colorectal cancer (CRC, 3rd), gastric cancer (5th), liver cancer (6th), esophageal cancer (8th), and pancreatic cancer (13th), based on GLOBOCAN 2018 data [1]. CRC, gastric, liver, esophageal, and pancreatic cancers also account for the 2nd, 3rd, 4th, 6th, and 7th leading cause of cancer-related deaths worldwide, respectively [1]. In addition to their prevalence and mortality rates, GI cancers pose major health issues of disparity between different sexes as well as different racial/ethnic groups, hence raising awareness of potential underlying socioeconomic issues and lifestyle/cultural differences.

Men are more likely to develop GI cancers than women in general. Compared to women, chances of men in developing esophageal cancer, HCC, gastric cancer, CRC and pancreatic cancer are 4.39-fold, 2.89-fold, 1.96-fold, 1.31-fold, and 1.30-fold higher, respectively, based on data collected from 2012–2016 (https://cancerstatisticscenter.cancer.org/, accessed on 30 November 2020) (Table 1). GI cancers also exhibit race/ethnic-related health disparities. Based on the CDC report, race is classified as non-Hispanic white (NHW), non-Hispanic black (NHB), Hispanic/Latino (H/L), Asian/Pacific Islander (A/PI), or American Indian/Alaska Native (AI/AN). Ethnicity is classified as Hispanic (which encompasses tremendous genetic diversity with ancestral contributions from Europeans, Africans, and Asians), or non-Hispanic [2]. Esophageal adenocarcinoma shows the most significant racial/ethnic disparity among GI malignancies between 2013 and 2017, with an incidence 5.6-fold higher in NHWs (3.4 per 100,000 person-years) than in A/PIs (0.6 per 100,000 person-years) and is approximately 3 times higher in NHWs than in other racial/ethnic groups (1.0–1.4 per 100,000 person-years). Conversely, esophageal SCC has the highest incidence in NHBs (2.7 per 100,000 person-years), compared to its incidence in other racial/ethnic groups at around 0.7–2.1 per 100,000 person-years. According to the National Surveillance by the North American Association of Central Cancer Registries 2012–2016 (https://cancerstatisticscenter.cancer.org/, accessed on 30 November 2020) and the Surveillance, Epidemiology, and End Results (SEER) databases and statistics 2013–2017 by the National Cancer Institute (https://seer.cancer.gov/explorer/, accessed on 30 November 2020), HCC shows the second most significant racial/ethnic disparity in both incidence and mortality among all GI cancers. Of all racial/ethnic groups in the United States, AI/ANs have the highest average annual liver cancer incidence, 2.2 times higher than NHWs who have the lowest incidence. The incidences of HCC in other racial/ethnic groups (NHB, A/PI and H/L) are close to that of AI/AN, ranging between 10.9 to 13.4 per 100,000 person-years. The incidence of cardia gastric cancer is higher in NHWs than in NHBs, and the incidence of non-cardiac gastric cancer is 3-fold higher in NHBs, A/PIs, and AI/ANs compared to NHWs. In contrast, NHBs show a higher CRC incidence than do A/PIs, Hispanics, and NHWs, as well as a higher CRC mortality rate, compared to A/PIs, Hispanics, and NHWs. Pancreatic cancer has a slightly higher incidence in NHBs than in A/PIs or other ethnic groups (15.9 vs. 9.3–12.7 per 100,000 person-years) (Table 1).

## 2. Liver Cancer, a Rising Malignancy with Sex and Race Disparities

Liver cancer is one of the few malignancies whose case numbers continue to rise despite recent diagnostic and therapeutic advances. HCC is the most common type of liver cancer (5th in men and 9th in women), making up 70–80% of all liver cases and accounting for almost half of all cancers in Southeast Asia [3]. Most HCC cases arise from precursor conditions, such as chronic viral hepatitis, cirrhosis, non-alcoholic fatty liver disease (NAFLD)/non-alcoholic steatohepatitis (NASH), and alcoholic liver disease (ALD). Less common etiologies for HCC include cigarette smoking, hemochromatosis, Wilson’s disease, aflatoxin B, and autoimmune hepatitis. In Southeast Asia, viral hepatitis (HBV and/or HCV) is the number one cause of HCC and the main driving force of its high prevalence. Chronic viral hepatitis accounts for 60%~80% of all cirrhotic cases in the region, followed by ALD and aflatoxin. In the past, global epidemiology of HCC has been largely determined by the prevalence of viral hepatitis and its age of contraction. However, with lifestyle changes and the development of screening and vaccination for HBV since 1982 and the implementation of immunotherapies for HCV (effective for 50–80% of the patients) since 2014, there is a global trend of shifting HCC etiology from viral hepatitis to NAFLD/NASH, particularly in industrialized countries [3,4]. In the US, it is estimated that, among US Medicare recipients from 2000 to 2011, metabolic disorders made up the largest component of HCC patients (32%), followed by HCV infection (21%), alcohol consumption (13%), cigarette smoking (9%), HBV infection (4%), and genetic disorders (1.5%) [5]. NAFLD is related to obesity and type 2 diabetes. ALD is caused by chronic consumption of alcohol and, similar to NAFLD, comprises a wide range of diseases from fatty liver, alcoholic hepatitis, liver fibrosis, to chronic cirrhosis, and finally leading to HCC. The prevalence rate of HBV infection in the US is much lower and accounts for much less of its total liver cancer cases (0.3–0.4% since the 1990s), compared to countries in Southeast Asia [6,7,8]. Finally, several known environmental toxins and chemicals are associated with HCC, such as aflatoxin B1, vinyl chloride, and androgenic steroids. Teasing out the differential impact of various HCC risk factors on males versus females or different racial groups may also illuminate the striking disparities in HCC incidence.

## 3. Epigenetic Changes: An Early Event in Oncogenesis

Epigenetic modifications play important roles in regulating gene expression and maintaining genome stability by changing the genomic structure without altering its underlying DNA sequence. It has emerged as a key mechanism that captures the early impact of environmental insults on the genome and may provide a missing piece of the puzzle for understanding the intersection between environmental inputs and genetic reactions during the progression of complex diseases, such as cancers, metabolic disorders, cardiovascular diseases, and neurodegenerative diseases. Increasing evidence emerging over the last 5–10 years suggests that alterations in the DNA methylation machinery are among the most common changes associated with neoplasia and may play a causative role in the early stages of carcinogenesis [9]. It is also worth noting that cancer-related DNA methylation changes may be widespread and shared by different tumor cell colonies as well as normal-appearing tissues that have not yet undergone histologically obvious morphological transformation [10]. In contrast, the majority of genetic mutations occur randomly as bystander events, and only a few act as true tumorigenic drivers [11]. Therefore, defining genetic mutations may provide insight into tumor heterogeneity and evolution but may be missing some of the early disease-driving events. More importantly, unlike genetic mutations, epigenetic modifications are dynamic and can potentially be reversed by therapies or environmental changes. Therefore, epigenetic research may discover new molecular targets that can be used to assess the risk of premalignant lesions developing into cancers (thereby allowing implementation of preventative measures), to detect cancers at an early stage when attempted curative therapy is still amenable, and/or to predict the response of cancers to therapeutic interventions.

There are three major mechanisms of epigenetic modification: DNA methylation, histone modification, and microRNA-mediated gene silencing. DNA methylation involves the addition of a methyl group to the 5′ position of the cytosine pyrimidine ring of a CpG dinucleotide, using S-adenosyl-methionine (SAM) as the methyl donor. Active demethylation of methylated cytosine involves hydroxylmethylation by Ten-eleven translocation (TET) enzymes, including Tet 1, Tet 2, and Tet 3, which sequentially convert 5-methylcytosine (5-mC) first to 5-hyroxymethylcytosine (5-hmC), then to 5-formylcytosine and finally to 5-carboxycytosine [12]. In an alternative pathway, 5-mC and 5-hmC are converted to thymine or 5-hydroxymethyluracil (5-hmU) through oxidative deamination. These intermediate bases result in G:X mismatches, where X is 5fC/5-caC/5-hmU/Thymine. Thymine-DNA glycosylase (TDG) excises the mismatched base from these G:X mismatches [13]. The resulting abasic site is then repaired by the base excision repair (BER) pathway to regenerate cytosine [14] (Figure 1). The enzymatic activities catalyzing DNA methylation can be classified into two types. The first type, which methylates the unmethylated cytosine residues on the hemi-methylated DNA strand after DNA replication, is catalyzed by DNA methyltransferase 1 (DNMT1) and its cofactor, ubiquitin-like with PHD and RING finger domains 1 (UHRF1) [15]. The second type, which adds new methyl groups to unmethylated cytosines, is catalyzed by two de novo DNA methyltransferases, DNMT3A and DNMT3B, together with their coactivator DNMT3L [16] (Figure 2). DNMT3A and DNMT3B are most needed during gametogenesis and early embryogenesis, when DNA methylation is erased and re-established in a genome-wide manner. The methylation pattern, once re-established, needs to be precisely maintained in order to sustain cell type-specific fates and functions. For this reason, the activity of de novo DNA methylation is turned off in most differentiated cells.

Histone modification also plays an important role in epigenetic regulation. It is a reversible process that regulates gene expression by switching between chromatin conformations that are favorable or unfavorable for gene transcription. Histone methylation and acetylation occur primarily on various lysine residues in the tail and globular domains of histones. The overall levels of these modifications are specifically monitored by “readers” that recognize the modified lysine in a sequence-dependent manner and precisely regulated by the activities of “writers” (methyltransferases and acetyltransferases) and “erasers” (demethylases and deacetylases) [17,18]. Acetylation is one of the most well characterized histone modifications. Histone acetylation is catalyzed by histone acetyltransferases (HATs) and is associated with an open chromatin state (euchromatin) with increased transcription factor-DNA interactions and gene expression [19]. Conversely, histone deacetylation is catalyzed by histone deacetylases (HDACs) and is associated with DNA condensation into heterochromatin, resulting in gene suppression [20]. Other histone modifications include phosphorylation, ubiquitylation, SUMOylation, ADP-ribosylation, crotonylation, hydroxylation, and proline-isomerization [21]. The third major epigenetic mechanism is regulated by microRNAs (miRNAs)—small non-coding RNAs of approximately 22 nucleotides in length. microRNAs function as negative gene regulators by base pairing with complementary sequences in the 3′ UTR of target mRNAs. Targeted mRNAs are silenced through RNA cleavage, destabilization of the poly-A tail, or interference with ribosomal translation [22].

CpG dinucleotides are widely and nonuniformly distributed throughout the human genome. Clusters of CpG dinucleotides, known as CpG islands (CGIs), are preferentially found in certain genomic regions. Approximately 45,000 CGIs are located close to promoter regions and have been functionally linked to gene expression regulation [23,24]. Aberrant DNA methylation may promote cancer progression via several mechanisms. First, promoter hypermethylation can silence the expression of tumor suppressor genes (TSGs), whereas promoter hypomethylation can lead to increased expression of oncogenes. It has been shown that the p16^INK4A^, Rb, and von Hippel-Lindau (VHL) tumor suppressors can be silenced by aberrant DNA hypermethylation in their promoter regions [25,26,27,28,29]. Conversely, DNA hypomethylation of RAS oncogene has been reported in HCC [30]. Second, global DNA hypomethylation is often seen before the formation of adenomas and continues to be maintained in neoplastic tissue, suggesting that changes in the DNA methylation landscape per se may contribute to neoplastic transformation [31]. DNA hypomethylation on transposable element repetitive sequences, such as short interspersed nuclear elements (SINEs or Alu elements) or long interspersed nuclear elements (LINEs), may predispose cells to chromosomal defects and rearrangements, leading to genetic instability and oncogenesis [24]. Third, DNA methylation also increases the probability of C-to-T mutations as a result of spontaneous hydrolytic deamination of 5-methylcytosine in CpG sites (Figure 1). Studies confirmed that CpG dinucleotides are mutation hotspots in a variety of genes [32,33,34,35,36]. A large number of mutations are found at methylated CpG dinucleotides in p53, and the five most commonly mutated codons in p53 (175, 245, 248, 273, and 282) contain methylated CpG dinucleotides [37]. Together, these data support that changes in DNA methylation profiles in neoplastic tissues may provide key information towards understanding and addressing sex and race/ethnic-related disparities in cancer incidence and mortality rates.

## 4. Literature Search on Sex and Race/Ethnic-Related DNA Methylation in HCC

To review the current knowledge on sex and race/ethnic-related changes in DNA methylation in HCC, PubMed searches were conducted to identify relevant publications from 1 January 1990 to 3 March 2021. To broaden the scope, only two key words (DNA methylation and liver cancer) were used in the initial PubMed search, and 1799 studies were identified (Figure 3). For sex-related disparities, abstracts of those studies were searched using the terms: sex, gender, women, female, or disparity, to identify potentially relevant studies. For race/ethnicity-related disparities, the following search terms were used: race, racial, ethnicity, ethnic, Asia, Asian, Caucasian, Europe, European, Africa, African, Hispanic, Whites, Black, or disparity. Articles identified by those search criteria and written in English were manually reviewed to further identify studies that reported regions or genes differentially methylated in HCC tumor tissues between males and females or different racial/ethnic or geographic populations. For sex/gender disparities, articles were excluded if they: (1) examined DNA methylation changes in plasma/serum samples but not in HCC tissue, and (2) did not perform statistical analysis of the difference between men and women. For race/ethnicity disparities, articles were included if: (1) studied populations were of predominantly monoethnic regions, such as China, Korea, Japan, and Europe, and (2) the racial and ethnic composition of the studied populations were clearly stated. Identified studies were all published after 2000, without a significant increase in the numbers of papers over time. Some papers on sex-related genetic variants that affect the DNA methylation status were also included [38,39,40].

Most studies applied methylation-specific PCR, restriction enzyme digestion, or bisulfite pyrosequencing in combinations for targeted genomic regions. Of the identified studies, eleven used methylation-specific PCR or quantitative real-time PCR, five used methylation-specific restriction enzyme digestion, and two used bisulfite pyrosequencing. Quantitative methylated DNA sequencing by pyrosequencing measures the level of methylated DNA at single nucleotide resolution and has been used to determine aberrant methylation levels in cancers [41]. Three of the studies examined methylation of the repetitive sequence Long Interspersed Nuclear Elements 1 (LINE1), with or without examining another type of short repeat, ALU. Repetitive sequences are widely distributed across the genome, thereby allowing assessment of the DNA methylation status across multiple genomic sites through the analysis of one specific sequence. LINE1 is a repetitive class I transposable element. As LINE1 occupies approximately 17% of the human genome, the extent of its methylation can be used as a surrogate marker of global DNA methylation. Notably, while the majority of these sites are truncated and inactive, about 80–100 LINE1 sites retain the ability to retrotranspose and therefore may contribute to genomic instability and evolution. One study reported LINE1 hypomethylation as a biomarker for poor prognosis in various types of gastrointestinal cancers [24].

Furthermore, many studies employed the Illumina Infinium Human Methylation 27 or 450K BeadChip technology to study DNA methylation in liver cancer in a genome-wide fashion. Only two analyzed the data based on sex or race [42,43]. The 27 BeadChip probes 27,578 individual CpG sites that are spread across 14,495 genes. The 450K BeadChip uses a patented bead chip technology that combines two different chemical assays, the Infinium I and Infinium II assays, to probe human DNA methylation at more than 480,000 cytosine sites [44], which account for ~1.6% of total human genomic CpG sites. With the advancement of Next-Generation Sequencing (NGS) technology, whole-genome bisulfite sequence (WGBS) has become the gold standard for genome-wide DNA methylation profiling. Our search identified 24 whole-genome DNA bisulfite sequencing studies of HCC, none of which analyzed the results based on sex/gender or race/ethnicity.

## 5. Sex-Dependent DNA Methylation Events in HCC

### 5.1. p16^INK4A^ and CDKL2: The Cyclin-Dependent Kinase-RB Pathway

The cyclin-dependent kinase (CDK) pathway contains multiple complexes formed by cyclins and their associated kinases in a step-wise manner. This pathway, along with its inhibitors (e.g., p16^INK4A^) and other regulators control the orderly progression of the cell cycle. Hypermethylation of the CDKN2A (Cyclin dependent kinase inhibitor 2A, encoding p16^INK4A^ and p14^ARF^), CCND2 (Cyclin D2, controlling G1/S progression), and CDKN2B (encoding p15^INK4B^) promoters is associated with the development of HCC (Figure 4) [45,46,47,48]. Li et al. reported a high incidence of p16^INK4A^ hypermethylation (58%) in HCC tissues, as compared to adjacent chronic hepatitis and cirrhosis tissue (16%) [49]. Another study also showed that p16^INK4A^ promoter hypermethylation frequencies were gradually elevated from 6% in normal liver tissue to 21% in adjacent non-tumor tissue to 41% in HCC tissue [50]. These observations suggest that p16^INK4A^ hypermethylation may contribute to the early events of hepatocarcinogenesis and persist throughout the process of malignant transformation. Wang et al. found that p16^INK4A^ promoter hypermethylation is significantly higher in men (46.9%) compared to women (27.2%) [50]. This is in conflict with the study by Li and co-workers, which reported a much higher frequency of aberrant p16^INK4A^ promoter methylation in women (83%) compared to men (50%), though this study is limited by the small number of female HCC patients [49]. While Wang and co-workers analyzed 118 patients (81 Men, 37 women), the Li study examined only 50 cases (38 men, 12 women). Given the conflicting findings, whether p16^INK4A^ promoter methylation contributes to the male-predominant HCC incidence remains inconclusive. Notably, gender differences in the methylation of the p16^INK4A^ promoter are observed not only in HCC but also in CRC [51]. The p16^INK4A^ gene is an endogenous CDK4/6 inhibitor. Palbociclib and Ribociclib are two FDA-approved CDK4/6 inhibitors for cancer treatment. One study showed that p16^INK4A^ methylation increased tumor sensitivity to the CDK4/6 inhibitor Palbociclib [52], suggesting that this might be an example of oncogene addiction. If sex-based differences in p16^INK4A^ promoter methylation are validated, this may be an avenue for sex-selective therapeutic protocols.

Cyclin-dependent kinase-like 2 (CDKL2), a new member of a separate branch of the CDK family, has decreased expression in several types of tumors [53,54]. CDKL2 methylation is dramatically increased in HCC tissue, in association with its decreased mRNA expression level [53]. Analysis of clinicopathological factors revealed that promoter hypermethylation of CDKL2 is significantly higher in women (48.0%) than in men (37.7%) (*p* = 0.037), suggesting that CDKL2 promoter hypermethylation may be a contributing factor to the sex disparity in HCC. These results indicate that aberrant DNA methylation may dysregulate the expression of genes that control cell cycle progression and account for one of the mechanisms underlying sex/gender-related disparity in HCC.

### 5.2. PPP2R1A and DNMT3B: Interaction between Methylation and Genetic Mutation

Protein phosphatase 2A (PP2A) is a major cellular phosphatase responsible for deactivating a variety of cellular signaling pathways. Chen et al. identified a genetic variant, known as rs11453459 or −241 (−/G), an insertion of guanine in the protein phosphatase 2 scaffold subunit α gene (PPP2R1A) promoter region located 241 bp upstream of its transcription start site (TSS) [40]. Their study showed that this DNA genetic variant was associated with gene silencing and this effect seemed less under the promoter methylation condition. In addition, the authors claimed that the −241 (−/G) insertion was more protective against HCC in young women than in young men (≤40 years of age). However, upon closer analysis, this appears to be due to the much higher frequency of the (−/G) allele in their female control group (38/46, 82.6%) compared to their male controls (−/G in 85/206, 41.3%), rather than the frequencies of the (−/G) allele in male (59/217 or 27.2%) versus female (11/34 or 32.4%) HCC patients. Thus, there is insufficient evidence at present to conclude that this variation contributes to sex differences in HCC incidence.

The DNMT3B gene has a C-to-T polymorphism at its promoter region (−149 bp), which renders a 30% increase in promoter activity and hence its gene expression level. Female HCC patients with the DNMT3B variant (−149 T) showed a much higher risk in developing HCC (OR (odds ratio): 1.97) compared to male patients (OR: 0.46), suggesting a possible genetic cause for the sex/gender disparity in HCC incidence, which may act by affecting the global DNA methylation status [39]. However, these results would need to be validated in a larger study, given the few cases observed (only 6 male (out of 58) and 8 female (out of 38) HCC patients had the −149 T variant). In addition, there remains a question of how large a population attributable fraction this variant would account for, even if the reported ORs are validated. Given the low apparent prevalence of this variant, other environmental exposures as well as genetic and/or epigenetic changes must account for the preponderance of sex-based HCC disparity, which sees a much higher incidence in males.

### 5.3. LINE1-Based Methylation Profiling

Retrotransposable elements (also known as “jumping genes”) are estimated to comprise 46% of the human genome and have been implicated in a variety of diseases including cancer [55]. LINE1 is a type of repetitive retrotransposon that may be up to several kb in length. LINE1 consists of a 5′ untranslated region containing a bi-directional internal promoter for RNA polymerase II with a CpG-rich region, two open reading frames (ORFs), and a 3′ terminal polyadenylation site. ORF1 encodes an RNA binding protein with a preference for cis binding LINE1 mRNA. ORF2 encodes a protein with both reverse transcriptase and DNA endonuclease activity that are essential for genetic transposition of LINE1 (or other retrotransposable elements at a reduced frequency). Short Interspersed Nuclear Elements (SINEs) are another class of repetitive retrotransposons approximately 80–400 bp in length that contain an internal promoter for RNA polymerase III and a 3′ A-rich tract. SINEs require activities encoded by other autonomous retrotransposons or the host for their mobility. The most common SINE is the human Alu sequence. Aberrant DNA methylation of LINEs and SINEs has been associated with increased genetic mutation and genomic instability—two common features of tumorigenesis [55]. Several studies have demonstrated that LINE1 is hypomethylated in HCC samples compared to paired adjacent normal tissues [56,57]. One study examined the average methylation status of three CpG sites in the LINE1 promoter region in 75 HCC patients. LINE1 hypomethylation was associated with significantly shorter disease-free survival (DFS) and overall survival (OS) rates. Furthermore, they found that sex is a significant factor affecting the methylation level of LINE1 promoter CpG islands in tumors. More HCC tumor samples from male patients (59%) showed LINE1 hypomethylation relative to the set of 75 HCC tumor samples than female patients (26%) [57], indicating that the extent of LINE1 demethylation in HCC tumors from male patients tends to be higher than that from female patients. However, all tumor samples (male and female) showed hypomethylation of LINE1 relative to adjacent non-tumor tissue, suggesting it is a common event in HCC tumorigenesis. Another group replicated the Zhu et al. finding of decreased LINE1 methylation in HCC compared to normal liver tissue but found no significant correlation between the methylation levels of LINE1 and Alu of different genders [58], leaving open the question of whether upregulation of LINE1 by demethylation contributes specifically to sexual disparities in HCC incidence. Therefore, while LINE1 hypomethylation may well play a role in genomic instability and initiation of HCC, strong evidence suggesting a role in sex-based HCC disparity is lacking at present (Figure 4).

Methyltetrahydrofolate reductase (MTHFR) is the key enzyme that catalyzes the final step of 5-methyl tetrahydrofolate (5-MTHF) synthesis from folic acid. 5-MTHF acts as the methyl donor for the synthesis of S-adenosyl methionine (SAM), the universal methyl donor for DNA methylation (Figure 4). One study showed that a C677T polymorphism in MTHFR reduced its enzymatic activity and that individuals with homozygous C677T showed DNA hypomethylation [59]. Notably, Qiao et al. reported that the MTHFR C677T variant is associated with lower LINE1 methylation, more so in women than in men, and that women have less LINE1 methylation compared to men in both chronic HBV and HCC, suggesting that LINE1 hypomethylation may be a more relevant pathogenic mechanism in female HCC patients with these MTHFR variants than in male patients [60].

### 5.4. Tumor-Related Gene Panel

One study examined DNA hypermethylation of seven tumor-related genes, including APC, WIF1, RUNX3, DLC1, SFRP1, DKK, and E-cad, in HCC patients, and showed that the hypermethylation status of those seven genes in HCC tissues, as defined by extent of CpG island methylation, differs significantly between male and female patients. The percentage of male patients harboring ≥ 3 hypermethylated genes (65%) is higher than that of female patients (41%) [61]. Another study examined the methylation status of CpG islands associated with 19 genes found to be hypermethylated in HCC compared to normal tissue, including HOXA1, CDKN1C, CRABP1, DLEC1, p16^INK4A^, CCND2, CACNA1G, RUNX3, PTGS2, BCL2, GRIN2B, NEUROG1, GSTP1, SYK, SFRP1, CALCA, SOCS3, APC, and TERT. Contrarily, their results showed that female HCC tissues have a higher number of hypermethylated genes (11.2 out of 19) than do male samples (8.4 out of 19) [58]. This could be interpreted to suggest that females require the inactivation of more TSGs than males for HCC development. However, neither study went on to examine the actual effect of DNA methylation on gene expression, leaving a list of candidate tumor-related genes with sex-based methylation differences in HCC for further evaluation of individual genes’ contribution to the sex/gender disparity of HCC in follow-up studies.

### 5.5. Genome-Wide Studies

To date, there is no study that directly examined HCC-related epigenetic differences between men and women using a genome-wide approach. One study by Shen et al. used the 450K microchip technology to identify differentially methylated CpG dinucleotides (DMCs) between HCV-associated HCC tumor tissue and adjacent normal tissue (ANT) [42]. They first identified 2441 hyper-DMCs and 127 hypo-DMCs in HCC tissues. Interestingly, 75 out of the top 500 DMCs show significant association with sex/gender but not with other risk factors. Some of the 75 DMCs and their associated genes may contribute to an epigenetic mechanism underlying the sex-dependent disparity in HCC incidence and outcome.

### 5.6. Additional Sex-Related DNA Methylation Changes

Aberrant DNA methylation in HCC was reviewed by Nishida et al. [47]. One gene differentially methylated in male and female HCC patients is the tumor necrosis factor (TNF) receptor superfamily member 12A (TNFRSF12A), encoding the Fn14 protein. Hypomethylation of two specific sites on the TNFRSF12A gene can distinguish HCC and alcoholic cirrhosis from other liver diseases. Stratification analysis revealed that hypomethylation of the CpG site (cg26808293) is associated with a poor prognosis, especially in male patients with an alcohol abuse history [62]. Glycine N-methyltransferase (GNMT) is a one-carbon group methyltransferase that regulates global DNA methylation by controlling the ratio of S-adenosylmethionine (SAM) to S-adenosylhomocystine. A study on GNMT knockout mice showed that genes of the mitogen-activated protein kinase (MAPK) pathway were activated in *Gnmt^−/−^* mice, which was especially striking in the female mice. Therefore, as a tumor suppressor gene for liver cancer, *GNMT* might be associated with gender disparity in liver cancer susceptibility through its influence on DNA methylation [38]. Suppressor of cytokine signaling 1 (SOCS1) is a suppressor of liver fibrosis and hepatitis-induced tumorigenesis [63], and SOCS1 hypermethylation is more likely found in males (63%) than in females (41%) [64]. One study by Wu et al. reported that the incidence of basonuclin 1 (BNC1, a zinc finger protein with unknown function) hypermethylation in tumor tissue was higher in female HCC patients (65%) compared to males (43%) (Figure 4) [65].

### 5.7. Sex-Dependent DNA Methylation Changes in Premalignant Liver Conditions vs. HCC

Different HCC risk factors show different prevalence rates between men and women, which may also contribute to apparent sex-related differences in HCC incidence. It has been reported that the prevalence rates for chronic HBV and HCV infection are over two times and 50% higher, respectively, in men (10.7% for HBV and 0.19% for HCV) compared to women (4.4% for HBV and 0.13% for HCV) in Southeast Asia [66,67]. In the US, incidences of HBV and HCV are also higher in men (3.5/10,000 for HBV and 0.8/100,000 for HCV) than in women (2.0/100,000 for HBV and 0.44/10,000 for HCV) (2016 US viral hepatitis surveillance; https://www.cdc.gov/hepatitis/statistics/2016surveillance/, accessed on 19 February 2021). Several groups have studied methylation changes linked to specific risk factors. For example, HCV infection was related to aberrant methylation of *SOCS-1*, *Gadd45β*, *MGMT*, *STAT1*, and *APC* [68]. On the other hand, HBV infection could affect DNA methylation of the *p16^INK4A^*, *GSTP1*, *CDH1*, *RASSF1A*, and *p21* genes, which may play important roles in the development of HCC [68], as well as affecting histone modification and microRNA expression [69]. Therefore, while a number of studies have observed differences in global methylation or that of specific gene promoters between male and female HCC patients, the study of epigenetic contributions to HCC sex disparities remains in its infancy with extensive need for validation and mechanistic analysis. A summary of the above-mentioned genes/pathways and studies is listed in Table 2.

## 6. Race-Dependent DNA Methylation Events in HCC

### 6.1. Wnt Pathway and Cell-Cell Adhesion

The Wnt/β-catenin signaling pathway is one of the most frequently activated signaling pathways in HCC [70,71]. It has been shown to play a role in the progression of chronic liver diseases to hepatocellular adenomas and early carcinomas [72]. β-catenin and adenomatosis polyposis coli (APC) are two key players in this pathway that play an oncogenic or tumor-suppressing role in various cancer types, including HCC (Figure 5) [73,74]. Promoter methylation has been recognized as an epigenetic mechanism regulating their gene expression during early stages of tumor development [73]. Hypermethylation of the APC promoter region was more common in HCC tissues compared to ANT (OR: 5.32) or normal liver tissues (OR: 20.43). Hypermethylation of the APC promoter is associated with an increased HCC risk more in A/PIs (OR: 29.7) than in NHWs (OR: 9.9) [75]. No study has yet compared the methylation status on the β-catenin gene among different racial and/or ethnic groups.

Another gene frequently lost during tumor progression in multiple cancer types is *CDH1* [76], which encodes the protein E-cadherin that maintains cell-cell contacts (Figure 5). Herath et al. examined the promoter methylation status of *CDH1* in HCC and ANT tissues from an Australian (76% Caucasian and 24% Asian) cohort and a South African (100% Black) cohort. *CDH1* promoter methylation was more prevalent in the Australian population compared to the South African population in the HCC tissues (30% in Australians and 13% in Africans) as well as in the ANT tissues (57% in Australians and 8% in Africans). However, it is not clear why the Australian cohort shows decreased *CDH1* promoter methylation in HCC compared to ANT, whereas the South African population shows the opposite. Furthermore, decreased E-cadherin protein levels were found in only one HCC sample in this study, suggesting that changes in E-cadherin DNA methylation may not play a direct role in driving the development of HCC via regulating its protein expression, at least in these populations [77].

### 6.2. The pRB and Cell Cycle Pathways

Herath et al. also examined the promoter methylation status of another six genes, including p16^INK4A^ (CDKN2A), p14^ARF^ (CDKN2A), and p15^INK4B^ (CDKN2B), using the methylation-specific PCR (MSP) approach and the same Australian and South African HCC and ANT samples as described above. They reported that the median MSP value of those six genes was significantly higher in the Australian cohort than in the South African cohort [78]. However, it is important to note that the median age between these cohorts was significantly different (35 years old for South Africans, 61 for Australians). Given that global methylation increases with age, one cannot rule out the possibility of age difference being a confounding factor. In another meta-analysis, subgroup analysis based on ethnicity revealed that methylation of p14^ARF^ was tightly associated with the risk of HCC in both the Chinese and Western populations (OR: 7.74 vs. 3.60). However, the difference in ORs was not directly compared between the Chinese and Western populations [79].

### 6.3. O^6^-MGMT DNA Repair Pathway

One of the genes examined by Herath and co-workers in the Australian (mixed Caucasian and Asian) and South African (Black) populations is O^6^-methylguanine DNA methyltransferase (O^6^-MGMT), which encodes an enzyme that repairs naturally occurring DNA lesions by converting O^6^-methylguanine to guanine. O^6^-MGMT is required to maintain genome stability. They did not detect methylation of O^6^-MGMT in any of their samples (Figure 5) [78]. Another meta-analysis study reported a weak (non-significant) association between O^6^-MGMT methylation and HCC risk in a Chinese population (OR: 2.42 with a 95% CI (confidence interval) of 0.76–7.73) but not in a Western population (OR: 0.06) [79], which supports that O^6^-MGMT methylation changes are unlikely to play a role in HCC disparities.

### 6.4. The GSTP1-Detoxification and SOCS1-JAK-STAT Pathways

Glutathione S-transferase (GST) is a family of enzymes responsible for detoxifying electrophilic compounds using glutathione, a key cellular antioxidant (Figure 5). The expression of GST pi isoform is regulated by promoter methylation in various tissues. The Suppressor of Cytokine Signaling 1 (SOCS1) protein regulates the inactivation of JAK-STAT signaling downstream of various cytokine and growth factor receptors. GSTP1 and SOCS1 are widely recognized as TSGs. Hypermethylation of the GSTP1 promoter region is present in most if not all human prostate cancers [80]. The abovementioned meta-analysis by Liu and co-workers also showed that the ORs of methylation on the GSTP1 and SOCS1 promoter regions are 18.8 and 13 in HCC tissues compared to normal liver tissues, respectively. Further subgroup analysis by ethnicity showed that promoter methylation of these two genes is specifically associated with HCC risk in Asians (OR: 34.7 (GSTP1), 29.0 (SOCS1)) and NHWs (OR: 8.9 (GSTP1), 5.3 (SOCS1)) [75]. Even though this study did not directly compare between different racial/ethnic groups, it provides two possible gene targets (GSTP1 and SOCS1), whose promoter methylation status may contribute to racial disparities in HCC incidence. Another study also found that hypermethylation of the GSTP1 promoter occurs less commonly in HCC in a European population (20.5%) [81] compared to the previously reported Asian population (65.1%) [82].

### 6.5. Additional Race/Ethnic-Related DNA Methylation Changes

The study by Herath et al. also used a combined bisulfide restriction analysis to compare the methylation status of five CpG islands (previously identified as hypermethylated in colorectal cancers) in HCC tumor tissues between an Australian cohort (74% Caucasian and 26% Asian) and a South African cohort (100% black) [78]. There are no genes associated with these five CpG islands as identified by a BLAST DNA homology search. This study reported a trend of higher methylation levels on these CpG islands in Australian HCCs compared to South African HCCs, even though the difference was not statistically significant. As mentioned above, one issue of the study is that there is a significant difference in age between the two cohorts (Median age: 61 vs. 35 years old), which might have confounded the methylation result, as age has been shown to increase the methylation level in general. To date, there is no study that directly examined HCC-related DNA methylation changes among different geographic populations or racial/ethnic groups using whole genome sequencing approaches. In addition to the sex-dependent difference in the MTHFR C677T polymorphism described above, it has also been reported that a Chinese population has a lower C677T frequency compared to a non-Asian population [83]. In this study, the authors reported a 30% reduction in the OR of HCC in the homozygous C677T population compared to the control population. However, this conclusion seems to be in conflict with the previous report showing a decreased survival associated with HCC LINE1 hypomethylation [57]. In addition, the same -149 T polymorphism of DNMT3B, which is associated with an increased risk of HCC development in a sex-dependent manner in a Moroccan population [39], also shows different prevalence in different populations, with the Moroccan population being the lowest (16% for homozygous TT), followed by an American NHW population (22%) [84] and a Chinese population (98%) [85]. Finally, one study utilized a genome-wide methylation assay (Illumina Methylation Epic BeadChip, 850K) to examine differentially methylated and differentially expressed (DMDE) genes in HCC vs. ANT tissue in European and African Americans. They identified 32 DMDE genes in European Americans and 40 DMDE genes in African Americans [43]. Surprisingly, only five DMDE genes were shared among these two populations.

### 6.6. Race-Related DNA Methylation Changes in Premalignant Liver Conditions vs. HCC

Similar to sex-related disparities in HCC, different racial/ethnic groups also vary in their prevalence rates of different HCC risk factors and/or etiologies. The prevalence of HCV infection in Asia and Africa is ~3.5%, which is much higher than that in Europe (<1%) and the United States (~1%). In line with the HCV infection prevalence, HCV infection-associated HCC cases are higher in Asia and Africa compared to other regions. In the United States, HCV had the largest population attributable fraction (PAF) among NHBs (36.1%) and Asians (29.7%), whereas metabolic disorders had the largest PAF among Hispanics (39.3%) and NHWs (34.8%) [5]. According to NHANES, NHBs in the US have a higher percentage of chronic HCV infection than do other races/ethnicities, and consequently HCV infection is also one of the most common HCC risk factors among this population. The prevalence rates of HBV infection also vary greatly among different racial/ethnic groups in the US, being the highest in Asians (3.1%), followed by NHBs (0.6%) and NHWs (0.1%). Because of the notable contribution of etiology to the race/ethnicity-associated disparities in prevalence, the apparent race/ethnic-dependent disparities in HCC may be secondarily driven by the different prevalence rates of risk factors among different racial/ethnic groups. In the US, obesity impacts NHBs more than Hispanics and less than NHWs, as indicated by statistics from the 2019 Adult Obesity Prevalence Maps by the CDC (https://www.cdc.gov/obesity/data/prevalence-maps.html, accessed on 26 February 2021). Therefore, to effectively analyze the possible primary contributions of DNA modification to racial/ethnicity HCC disparities, study designs need to take into account differences in HCC etiologies and risk factors present in the populations studied. At present, initial studies support a potential role of differential promoter methylation of APC, GSTP1, and SOCS in Asian HCC patients in possibly accounting for some of the racial group differences in HCC incidence. It remains to be determined whether these differences are truly oncogenic or mere passengers of broader genomic and epigenetic instability. Furthermore, whether these promoter methylation changes are the result of known HCC etiopathological conditions such as chronic HCV infection remains to be established. A summary of the above-mentioned genes/pathways and studies is listed in Table 3.

## 7. Future Research Trends

### 7.1. Common Confounding Factors in Health Disparity Analyses

The development of HCC is linked to well-identified etiological risk factors. The contribution of specific risk factors to HCC development varies greatly by geographic location. Currently, chronic HBV infection is the predominant risk factor for HCC in Southeast Asia and Africa, whereas chronic HCV infection is the main HCC risk factor in Japan, Latin America, the United States, and Western countries. In the United States, NAFLD/NASH make up the largest component of HCC patients, followed by HCV infection, heavy alcohol consumption, and cigarette smoking [5]. Exposure to the fungal metabolite aflatoxin B1 (which usually contaminates long-term stored grains, such as rice, corn, soybeans and peanuts) is one of the major risk factors of HCC due to dietary habits in some regions of the world, such as China, Africa, and Central/South America. Furthermore, in countries with large immigrant populations, the contribution of specific risk factors varies among different ethnic groups, as well as between locally born vs. foreign-born patients within a geographic location. In the United States, most patients who tested positive for HBV were foreign born. Non-Hispanic Asians have the highest HBV prevalence [86]. Different racial groups have different genetic and epigenetic make-ups, which may predispose a race to specific HCC risk factors (e.g., viral infection, obesity, type 2 diabetes, excess alcohol consumption, etc.). This fact of biased contribution of etiological risk factors in different populations makes the task of unveiling genetic and epigenetic contributions to racial predisposition extraordinarily difficult.

From a scientific standpoint, the concept of race itself is problematic, due to the lack of validated biological markers for racial assignment. While race plays a role in the self-identity of many individuals as a sociological concept, there can be a wide range of genetic/epigenetic variation within a given “race” as currently defined. Furthermore, the self-identification of individuals with a specific race may be at variance with their ancestral genetic heritage, which may, for instance, include contributions from multiple races. Currently, self-reported race/ethnicity is widely used in epidemiological studies as a record of genetic background origin. However, self-reported race/ethnicity may not accurately represent their genetic composition, especially in mixed populations, such as Hispanic, which are admixed with European, African, and Native American ancestry. Genetic composition varies greatly from individual to individual in mixed populations. From a scientific standpoint, race is defined better by germline (biological) ancestry than by self-identified ancestry, as genetic and epigenetic components may be mixed in a continuous spectrum within individuals self-identified as one race. Given the complexity in defining germline (biological) ancestry using objective measurements (e.g., polygenic race score) and that epi/genetic signatures may or may not exist to clearly define race, major challenges remain in the research of race-related health disparities.

### 7.2. Basic and Translational Research

Several questions remain unanswered regarding the role of methylation and other epigenetic changes in total HCC incidence, as well as in accounting for sex- and race-based HCC disparities. As mentioned above, apart from several meta-analyses on APC, GSTP1, SOCS, and P14, some of the identified methylation changes between groups require validation or have conflicting reports. In addition, it should be noted that studies finding no significant differences between different racial or sex groups, such as the papers examining O^6^-MGMT, may be limited by heterogeneity within the tumor samples obscuring genuine methylation differences between groups. In select cases, further probing of preliminary negative findings may be warranted. Confirming that methylation changes reflect shifts in gene expression, and assessing the oncogenic contribution (if any) of specific differentially methylated genes remain substantial tasks to be addressed in the near term.

One area that has been virtually untouched is examining gene methylation differences between sexes or racial groups in healthy individuals, and assessing whether these differences may result in divergent levels of risk from the same exposure. An initial step towards addressing these questions was done by McCormick and co-workers using animal models, who catalogued sex-based methylation differences between isogenic mice across multiple tissues [87]. Differences in baseline gene expression due to epigenetic variation between groups may generate differential susceptibility to a given risk factor such as smoking or HBV infection, although at present this remains speculative.

Furthermore, it is likely that substantial numbers of differentially methylated genes remain to be discovered. Whole genome bisulfite sequencing (WGBS) at the single nucleotide resolution level enhances the possibility of discovering novel predictive and prognostic markers, not limited by our prior knowledge. To understand the totality and dynamics of the disease, it may be necessary to do WGBS on a spectrum of patients representing both sexes and different racial/ethnic groups across a wide range of disease states. The potential findings will likely provide new perspectives on our understanding of HCC as well as new avenues for clinical intervention. In the past, most DNA methylation analyses were performed using tissue samples collected from precancerous and/or cancerous livers. The invasive nature of tissue biopsy poses a serious ethical issue that limits the source of available samples as well as the feasibility of applying the discovered DNA methylation markers as a diagnostic tool in the clinic. In recent years, liquid biopsy based on circulating cell-free DNA (cfDNA) has emerged as a new research, diagnostic, and/or screening platform to determine genetic and epigenetic changes in patients previously inaccessible due to ethical considerations [88,89]. Several circulating epigenetic markers are currently being evaluated as diagnostic markers for HCC, including GSTP1 [90], SEPT9 [91], RASSF1A, E-cadherin, and RUNX3 [92]. Liquid biopsy-based biomarker analysis promises to be a major research trend in the future. As in all discovery studies, the road from bench to bedside may be long, requiring careful validation using preclinical models and multi-phase clinical trials.

### 7.3. Clinical Importance

We foresee several clinically relevant outcomes from studies of HCC methylation differences between sexes and different racial groups. Knowledge of underlying epigenetic differences tied to differential sensitivity to given HCC risk factors can be used to inform public education and lifestyle intervention efforts in a preventive medicine context. These can also be used, in principle, in screening efforts to generate tailor-made individual risk assessments (personalized medicine). Furthermore, epigenetic changes linked to specific genes that are correlated with differential survival of HCC patients of different sexes or racial groups can hold diagnostic or prognostic value, and perhaps become incorporated in treatment as well as health policy decisions. Finally, development of therapeutics targeting key epigenetic changes hold great hope for overcoming several drawbacks of traditional cancer therapies. Since epigenetic changes are reversible (in contrast to genetic mutations), there is a higher chance that oncoepigenetic changes acquired over the course of tumor development may be fully reversed by epigenetic-targeting therapies. Furthermore, it may be possible to design epigenetic-targeting approaches that reduce tumor genomic instability, thereby reducing the tumor’s capability of adapting to a therapeutic cocktail by developing resistance. Finally, it is possible that epigenetic-targeting therapies may be less toxic to non-tumor tissues than traditional cancer therapeutics, reducing the severity or extent of treatment side effects.

## Figures and Tables

**Figure 1 ijms-22-03820-f001:**
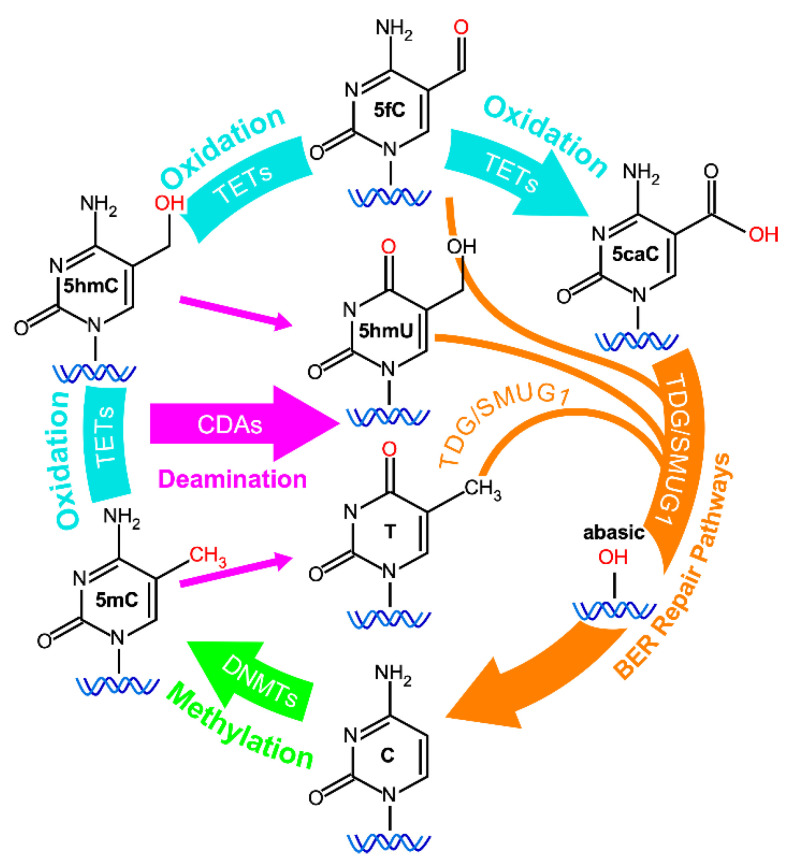
Cytosine methylation/demethylation cycle. Cytosine (C) can be modified by DNA methyltransferases (DNMT) at the 5 position to generate 5-methylcytosine (5mC). 5mC may be sequentially oxidized by Ten Eleven Translocation methylcytosine dioxygenase (TET) to generate first 5-hydroxymethylcytosine (5hmC), and then 5-formylcytosine (5fC) and 5-carboxycytosine (5caC). Alternatively, 5mC or 5hmC may be deaminated by cytidine deaminase (CDA) to generate thymidine (T) or 5-hydroxymethyluracil (5hmU), respectively. The base excision repair pathway (BER), along with thymine DNA glycosylase (TDG) and single-strand-selective monofunctional uracil-DNA glycosylase (SMUG1), is required for re-generation of cytosine.

**Figure 2 ijms-22-03820-f002:**
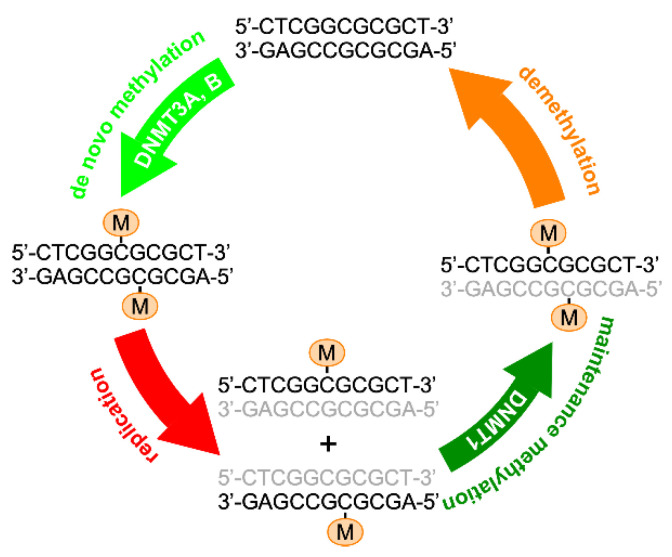
De novo versus maintenance DNA methylation enzymes. Unmethylated CpG sequences may undergo de novo cytosine 5-methylation, catalyzed by DNMT 3A and DNMT3B. DNA replication produces two copies of hemi-methylated double-stranded DNAs that are fully methylated by DNMT1 (maintenance methylation). Cytosine methylation may be removed through the demethylation process described in Figure 1. Grey letters and M denote newly synthesized DNA strands and the methyl group, respectively.

**Figure 3 ijms-22-03820-f003:**
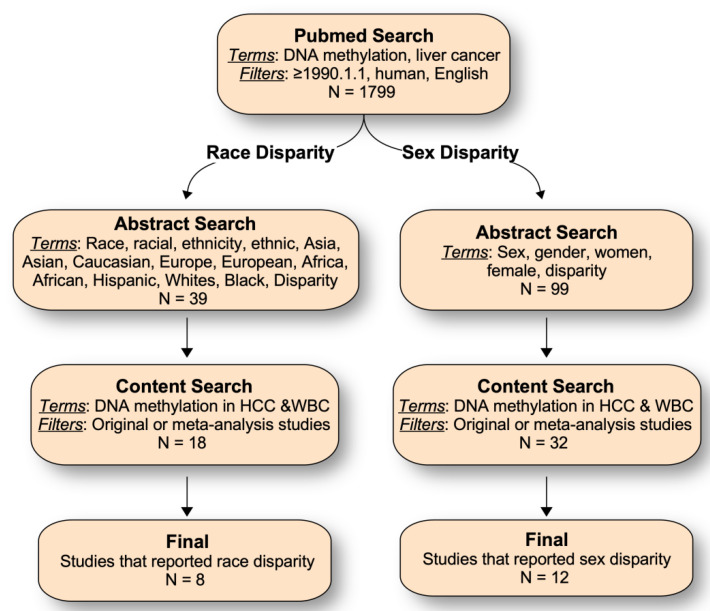
Schematic of literature search strategies used in this study.

**Figure 4 ijms-22-03820-f004:**
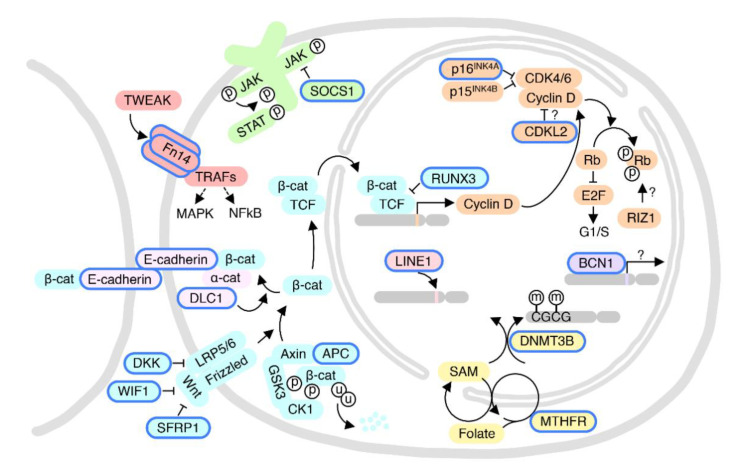
Cellular pathways implicated in sex/gender-related DNA methylation changes in HCC. Color schemes represent different pathways. Proteins regulated by sex/gender-related DNA methylation, or with gene variants having sex/gender-related differences, are highlighted by blue margins. Circles labeled with p, u, and m represent phosphorylation, ubiquitination, and methylation, respectively. Pointed and blunted arrows represent activation and inhibitory activities, respectively. Question marks indicate unclear underlying mechanisms. See text for protein abbreviations.

**Figure 5 ijms-22-03820-f005:**
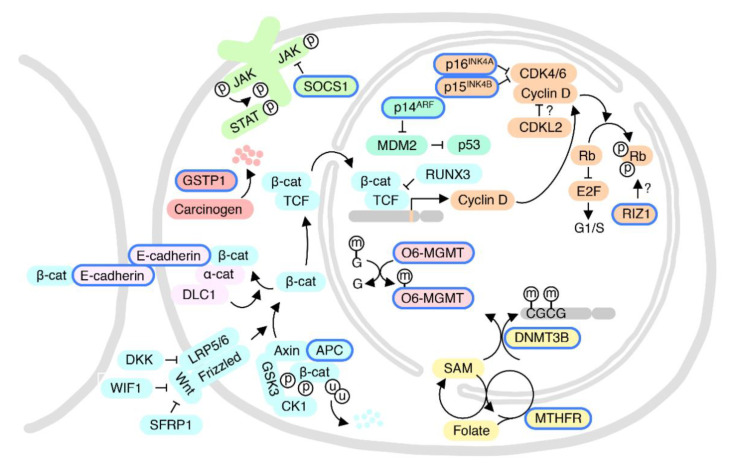
Cellular pathways implicated in race/ethnicity-related DNA methylation changes in HCC. Proteins regulated by race/ethnicity-related DNA methylation are highlighted by blue margins.

**Table 1 ijms-22-03820-t001:** Health disparities in gastrointestinal malignancies.

		Colorectal		Liver		Gastric		Esophagus		Pancreatic	
		Incidence	Mortality	Incidence	Mortality	Incidence	Mortality	Incidence	Mortality	Incidence	Mortality
Sex	M	44.4	16.6	12.7	9.6	9.0	4.1	7.9	7.0	14.6	12.7
	F	34.0	11.7	4.4	4.0	4.6	2.2	1.8	1.4	11.2	9.6
Race	NHW	38.6	13.8	6.9	5.8	5.4	2.3	5.0	4.4	12.7	11.1
	NHB	45.7	19.0	10.9	8.6	10.1	5.4	4.0	3.2	15.9	13.6
	H/L	34.1	11.1	13.4	9.3	9.6	5.0	2.7	2.0	11.2	8.5
	A/PI	30.0	9.5	12.7	9.0	10.3	5.1	2.1	1.6	9.3	7.5
	AI/AN	43.3	15.8	15.1	10.6	8.8	4.8	4.5	3.4	11.5	8.9

Incidence: Cancer Statistics Center, Age-Adjusted Rate per 100,000 person-year, 2012–2016; Mortality: Cancer Statistics Center, Age-Adjusted Rate per 100,000 person-year, 2013–2017; Abbreviations: M, male; F, female; NHW, non-Hispanic white; NHB, non-Hispanic black; H/L, Hispanic/Latino; A/PI, Asian/Pacific Islander; AI/AN, American Indian/Alaska Native.

**Table 2 ijms-22-03820-t002:** Genes or genomic loci showing sex-related differential methylation in hepatocellular carcinoma (HCC).

Reference	*Genes*	Results	Design and Method			
			Tissues	Etiology	Population	Method
Li X et al. [49]	*CDKN2A (p16^INK4A^)*	Hypermethylation (F > M)	Liver	Virus, Unknown	Japanese	MSP
Wang Y et al. [50]	*CDKN2A (p16^INK4A^)*	Hypermethylation (M > F)	Liver	Virus (HBV)	Chinese	MSRE-qPCR, qBS
Zhou Y et al. [53]	*CDKL2*	Hypermethylation (F > M)	Liver	Virus, Unknown	Chinese	MSRE-qPCR, qBS
Zhu C et al. [57]	*LINE1*	Hypomethylation (M > F)	Liver	Virus, Alcohol, Unknown	Japanese	qBS
Lee HS et al. [58]	*HOXA1, CDKN1C, CRABP1, DLEC1, CDKN2A (p16^INK4A^), CCND2, CACNA1G, RUNX3, PTGS2, BCL2, GRIN2B, NEUROG1, GSTP1, SYK, SFRP1, CALCA, SOCS3, APC* and *TERT*	Numbers of hypermethylated genes (F > M)	Liver	Virus (HBV)	Korean	qMSP
	*LINE1*	Hypomethylation (No sex difference)	Liver			MSRE-PCR
Liu JB et al. [61]	*APC, WIF1, RUNX3, DLC1, SFRP1, DKK, CDH1*	≥ 3 hypermethylated genes (M > F)	Liver	Virus, Unknown	Chinese	MSP
Shen W et al. [42]	75 DMCs	Significant sex difference (site-dependent)	Liver	Virus, Alcoholic, Tobacco	American	BeadChip (450K)
Wang Y et al. [62]	*TNFRSF12A*	Association of hypomethylation and poor outcome (M > F)	Liver	Virus, Alcohol, Unknown	American	BeadChip (450K, 27K)
Zhang X et al. [64]	*SOCS1*	Hypermethylation (M > F)	Liver	Virus (HBV)	Chinese	MSRE-qPCR
Wu Y et al. [65]	*BNC1*	Hypermethylation (F > M)	Liver	Virus, Unknown	Chinese	MSRE-qPCR
Ezzikouri S et al. [39]	*DNMT3B*	Higher HCC risk in *DNMT3B* -149T carriers (F > M)	Blood	Virus, Unknown	Moroccan	PCR-RFLP
Qiao K et al. [60]	*MTHFR*	LINE1 hypomethylation in *MTHFR* 677T carriers (F > M)	Blood	Virus (HBV)	Chinese	gPCR, qMSP

Gene symbols: refer to text for full names; DMCs: differentially methylated CpG dinucleotides; “Hypermethylation” indicates HCC shows increased methylation compared to normal; “Hypomethylation” indicates HCC shows decreased methylation compared to normal; MSP: methylation specific PCR; qMSP: quantitative methylation specific PCR; MSRE-qPCR: methylation-specific restriction enzyme-qPCR; MSRE-PCR: methylation-specific restriction enzyme-PCR; qBS: quantitative bisulfite sequencing; PCR-RFLP: PCR-restriction fragment length polymorphism; gPCR: genomic PCR.

**Table 3 ijms-22-03820-t003:** Genes or genomic loci showing race/ethnicity-related differential methylation in HCC.

Reference	Genes	Results	Design and Method			
			Tissue	Etiology	Population	Method
Liu M et al. [75]	*APC, GSTP, SOCS1*	Association of hypermethylation and HCC risk (Asian > Whites)	Liver	Unavailable	Asian, American, European	MSP, qMSP
Herald NI et al. [77]	*CDH1*	Hypermethylation (Caucasian/Asian > African Black)	Liver	Virus, Alcohol, Aflatoxin, Hemochromatosis, Allagille’s syndrom, Unknown	Australian, South African	MSP
Herald NI et al. [78]	*CDKN2A (p16^INK4A^), CDKN2A (p14^ARF^), CDKN2B (p15^INK4B^), CDH1, MGMT, RIZ1*	Hypermethylation (Caucasian/Asian > African Black)	Liver	Virus, Alcohol, Aflatoxin, Hemochromatosis, Allagille’s syndrom, Unknown	Australian, South African	MSP
	5 CpG islands	Hypermethylation (Caucasian/Asian > African Black)				MSRE-qPCR
Li CC et al. [79]	*CDKN2A (p14^ARF^), MGMT*	Association of hypermethylation and HCC risk (Chinese > Western)	Liver	Unavailable	Chinese, Japanese, Australian, South African, Spain, American, German	MSP
Anzola M et al. [81]	*SOCS1*	Hypermethylation (Asian > European)	Liver	Virus, Alcohol, Tobacco, Unknown	Spain	MSP
Varghese RS et al. [43]	40 DMDE genes in European American, 32 DMDE genes in African American	Only 5 genes overlapped in DMDE genetic profiles	Liver	Tobacco, Virus, Alcohol	American	BeadChip (850K)
Yuan JM et al. [83]	*MTHFR*	Frequency of MTHFR 677T allele carriers (non-Asian > Asian)	Blood	Tobacco, Alcohol, Virus	American, Chinese	gPCR
Ezzikouri S et al. [39]	*DNMT3B*	Frequency of DNMT3B -149T carriers (Chinese > Moroccan)	Blood	Virus, Unknown	Moroccan	PCR-RFLP

DMDE: Differentially methylated, differentially expressed genes that are targets of significantly expressed miRNAs in the population; Gene symbols and other abbreviations: refer to text and Table 2 for full names.

## Data Availability

Not applicable.

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
