# Peer review of "Sex and Race-Related DNA Methylation Changes in Hepatocellular Carcinoma"

_ijms, 2021, doi:10.3390/ijms22083820_

Round 1

Reviewer 1 Report

This is the nice review of hepatocellular carcinoma (HCC) focusing on sex and race-related DNA methylation changes. Explanations of basic knowledge of epigenetic changes and methylation-related genes / pathways are excellent. Their literature searching is strategic and reasonable. However, I have some comments.

  1. Although the each molecular-based explanation is excellent, results of literature searching, correlations of sex- or race-dependent methylations to HCC etiologies are hard to understand only by documentation. Please provide Table(s) explaining the genes / pathways detected by literature searching, sex/race predominance, and etiologies / study designs of HCC with relevant literatures.
  2. In Page 2, the authors described the etiologies of HCC, such as viral hepatitis, cirrhosis, NAFLD/NASH, alcohol, and uncommon etiologies. Cigarette smoking is also recognized as one of etiologies of HCC. Please mention to smoking as HCC etiology.
  3. In section 7 (future research trends), only documentations regarding HBV/HCV infection and exposure of aflatoxin B1 were found. Alcohol / cigarette consumption and rate / degree of obesity are expected for etiologies having difference in sex and/or race. Please mention to these matters.

Author Response

We thank both reviewers for their thoughtful comments on our manuscript. We have addressed all the critique by adding new text (see track changes) and tables (Table 2 and Table 3). I hope that the revised manuscript will be deemed responsive to the reviewers’ comments and suitable for publication in IJMS. My point-by-point responses to the reviewers’ comments are provided below.

Reviewer 1

  1. Although the each molecular-based explanation is excellent, results of literature searching, correlations of sex- or race-dependent methylations to HCC etiologies are hard to understand only by documentation. Please provide Table(s) explaining the genes / pathways detected by literature searching, sex/race predominance, and etiologies / study designs of HCC with relevant literatures.
  • Done (see new Table 2 and Table 3)
  1. In Page 2, the authors described the etiologies of HCC, such as viral hepatitis, cirrhosis, NAFLD/NASH, alcohol, and uncommon etiologies. Cigarette smoking is also recognized as one of etiologies of HCC. Please mention to smoking as HCC etiology.
  • Done (see page 2, line 73-74)
  1. In section 7 (future research trends), only documentations regarding HBV/HCV infection and exposure of aflatoxin B1 were found. Alcohol / cigarette consumption and rate / degree of obesity are expected for etiologies having difference in sex and/or race. Please mention to these matters.
  • Done (see page 15, line 600-602, 610-612)

Reviewer 2 Report

This review article indicated sex and race-related DNA methylation changes in hepatocellular carcinoma (HCC). HCC occurrence shows the significant sexual and racial/ethnic disparity, and it is very important to review current knowledge on sex and race-related DNA methylation changes in HCC. This article is well-written, but there are same problems.

  1. In Table 1, the authors should describe the spelling for abbreviations in the footnote.
  2. According to Figure 3, finally, studies that reported race disparity and that reported sex disparity were 8 and 12 reports, respectively. There were no details about 20 reports in this article. I consider that these findings should be shown in the table.
  3. It is very easy to understand this article for readers, if you create the table in which the differences among sexes or races for each DNA methylation of several genes are demonstrated.
  4. The differences among sex and race-related DNA methylation changes in HCC were almost analyzed in HCC tissues. Recently, liquid biopsy based on methylated DNA, such as CDKN2A, RASSF1A, GSTP1, and SEPT9, etc., has been suggested as a novel diagnostic tool for HCC. You should describe a foresight of this tool in the Future research trends section.

Author Response

We thank both reviewers for their thoughtful comments on our manuscript. We have addressed all the critique by adding new text (see track changes) and tables (Table 2 and Table 3). I hope that the revised manuscript will be deemed responsive to the reviewers’ comments and suitable for publication in IJMS. My point-by-point responses to the reviewers’ comments are provided below.

Reviewer 2

  1. In Table 1, the authors should describe the spelling for abbreviations in the footnote.
  • Done (see Table 1 footnote)
  1. According to Figure 3, finally, studies that reported race disparity and that reported sex disparity were 8 and 12 reports, respectively. There were no details about 20 reports in this article. I consider that these findings should be shown in the table.
  • Done (see new Table 2 and Table 3)
  1. It is very easy to understand this article for readers, if you create the table in which the differences among sexes or races for each DNA methylation of several genes are demonstrated.
  • Done (see new Table 2 and Table 3)
  1. The differences among sex and race-related DNA methylation changes in HCC were almost analyzed in HCC tissues. Recently, liquid biopsy based on methylated DNA, such as CDKN2A, RASSF1A, GSTP1, and SEPT9, etc., has been suggested as a novel diagnostic tool for HCC. You should describe a foresight of this tool in the Future research trends section.
  • Done (page 16, line 660-670)